# Glyoxalase-1-Dependent Methylglyoxal Depletion Sustains PD-L1 Expression in Metastatic Prostate Cancer Cells: A Novel Mechanism in Cancer Immunosurveillance Escape and a Potential Novel Target to Overcome PD-L1 Blockade Resistance

**DOI:** 10.3390/cancers13122965

**Published:** 2021-06-13

**Authors:** Cinzia Antognelli, Martina Mandarano, Enrico Prosperi, Angelo Sidoni, Vincenzo Nicola Talesa

**Affiliations:** 1Department of Medicine and Surgery, Bioscience and Medical Embryology Division, University of Perugia, L. Severi Square, 06129 Perugia, Italy; vincenzo.talesa@unipg.it; 2Section of Anatomic Pathology and Histology, Department of Medicine and Surgery, University of Perugia, L. Severi Square, 06129 Perugia, Italy; martina.mandarano@unipg.it (M.M.); enrico.prosperi@unipg.it (E.P.); angelo.sidoni@unipg.it (A.S.)

**Keywords:** glyoxalase 1, methylglyoxal, MG-H1, metastatic prostate cancer, PD-L1/PD-1, immune check-point, immunosurveillance, atezolizumab

## Abstract

**Simple Summary:**

Metastatic prostate cancer (mPCa) is a well-known lethal condition. One of the mechanisms through which PCa cells become so aggressive is the avoidance of immune surveillance that further fosters cell growth, invasion, and migration. PD-L1/PD-1 axis plays a crucial role in inhibiting cytotoxic T cells and maintaining an immunosuppressive cancer microenvironment. Hence, targeting PD-L1/PD-1 axis represents a potential way to control mPCa. Unfortunately, mPCa patients do not respond to PD-L1/PD-1 axis blockade, focusing the research to understand the possible underpinning mechanisms. Our results provide a novel pathway taking part in cancer immunosurveillance escape and in the above-mentioned immunotherapy resistance, which provides the basis for additional studies aimed at developing novel therapeutic opportunities, possibly also in combination with antibodies blocking PD-L1/PD-1 axis.

**Abstract:**

Metastatic prostate cancer (mPCa) is a disease for which to date there is not curative therapy. Even the recent and attractive immunotherapeutic approaches targeting PD-L1, an immune checkpoint protein which helps cancer cells to escape from immunosurveillance, have proved ineffective. A better understanding of the molecular mechanisms contributing to keep an immunosuppressive microenvironment associated with tumor progression and refractoriness to PD-L1 inhibitors is urgently needed. In the present study, by using gene silencing and specific activators or scavengers, we demonstrated, in mPCa cell models, that methylglyoxal (MG), a potent precursor of advanced glycation end products (AGEs), especially 5-hydro-5-methylimidazolone (MG-H1), and its metabolizing enzyme, glyoxalase 1 (Glo1), contribute to maintain an immunosuppressive microenvironment through MG-H1-mediated PD-L1 up-regulation and to promote cancer progression. Moreover, our findings suggest that this novel mechanism might be responsible, at least in part, of mPCa resistance to PD-L1 inhibitors, such as atezolizumab, and that targeting it may sensitize cells to this PD-L1 inhibitor. These findings provide novel insights into the mechanisms of mPCa immunosurveillance escape and help in providing the basis to foster in vivo research toward novel therapeutic strategies for immunotherapy of mPCa.

## 1. Introduction

Metastatic prostate cancer (mPCa) is a clinically relevant stage of PCa due to its high mortality rate [1,2]. Studies aimed at unveiling the molecular mechanisms underpinning the onset of mPCa have become a research field of utmost importance, representing still today an open challenge.

Multiple mechanisms underlying the development of mPCa have been described [3,4], most of which confined to cancer epithelial cells. However, it has become evident that a crucial role in mPCa genesis is also played by the tumor microenvironment, consisting of non-immune and immune cells [1]. In general, microenvironment-residing immune cells negatively control cancer development and progression by restraining tumor expansion or destroying tumors [5]. As other neoplastic diseases, PCa is generally considered a “cold cancer,” namely a low immune-reactive cancer since it presents either limited infiltration of immune cells or, conversely, extensive infiltration of immunosuppressive T cells [6,7,8,9] that, however, are tolerated and become incapable of mediating cancer cell destruction [10,11,12]. The suppressed tumor immune microenvironment and/or a defective cell-mediated immune response are important factors driving PCa tumor progression [12]. Among the different pathways involved in PCa immune surveillance escape, the immune check-point programmed death ligand 1 (PD-L1) and its receptor programmed cell death protein 1 (PD-1) has gained great attention. In general, the interaction between PD-L1, expressed in cancer cells, and PD-1, a T-cell membrane protein, inhibits CD8+ cytotoxic T-lymphocyte survival and proliferation and affects their function, ultimately suppressing the immune response and inducing immune tolerance of cancer [13]. PD-L1 is highly expressed in aggressive PCa where it assists antitumor immune escape, driving tumor proliferation and progression, promoting tumor recurrence and explaining primary therapy failure [14,15]. Building on these evidences, targeting PD-L1/PD1 axis has become an attractive immunotherapeutic strategy and, indeed, its clinical value has been confirmed in a number of human cancers, such as melanoma, lung cancer, and renal cell carcinoma, but not in PCa, mPCa included, except in a small cohort of selected patients and in combination with other standard therapies [16,17,18,19]. However, the very limited information available in mPCa does not allow the scientific community to draw definitive conclusions on the potential efficacy of therapeutic molecules directed at blocking PD-L1/PD1 axis.

In view of all this, focusing the research to understand the regulatory mechanisms of the PD-L1/PD-1 axis and, consequently, to develop novel therapeutic strategies to drive PD-1/PD-L1 blockade and/or help to overcome hurdles of targeting it, is an urgent need. Besides, although it is one of the most studied immune checkpoints, PD-L1/PD-1 pathway still presents several aspects of its biology that must be clarified [20].

Glyoxalase 1 (Glo1) is the first and rate-limiting catabolic enzyme in the degradation pathway of methylglyoxal (MG), a highly reactive intermediate of glycolysis, known as the major precursor of advanced glycation end products (AGEs) [21]. The role of Glo1/MG-AGEs axis in promoting PCa progression is by now ascertained [21]. Glo1, is overexpressed in advanced mPCa, where, through specific MG-derived AGEs, it acts as a determinant of cancer survival [22,23] and metastatic behavior [24,25]. It remains unknown, however, whether Glo1 expression is specifically used by mPCa also as a mechanism of immune evasion. Although never investigated before, it seems that a role in this ambit is very plausible. In fact, it has been recently demonstrated that tumors, primarily dependent on glycolysis to support their elevated growth demands (Warburg effect) [26], like typically mPCa is [27], are able to modulate expression of specific genes relevant to cancer cell immune escape [28]. Of note, Glo1 is an enzyme involved in glucose metabolism due to its capacity of metabolizing MG, a by-product of glycolysis through the intermediates, in particular, from the non-enzymatic degradation of dihydroxyacetone phosphate and glyceraldehyde-3-phosphate [21]. In addition, it has been very recently observed that the expression of PD-L1 is attenuated in Glo1-knockout malignant melanoma cells [29], thus suggesting a potential causative role of Glo1 in controlling PD-L1 expression in this neoplasia, and possibly also in mPCa. As to MG, its immune regulatory roles have been already previously documented in different settings, even though conflicting results have been reported. If on the one side MG would impair immune function [30,31,32,33] or do not affect lymphocytes viability [34], on the other hand it would stimulate the immune system against tumor cells [35,36,37]. To the best of our knowledge, the immune modulatory role of MG in PCa has never been investigated. 

In the present study, we first evaluated whether and how Glo1/MG axis might be used by mPCa as a novel mechanism of immune evasion, focusing on PD-L1. Second, we wanted to investigate whether Glo1/MG pathway might be involved in the refractoriness of this neoplasia to PD-L1/PD-1-based immunotherapy, with a particular interest on the PD-L1-directed monoclonal antibody, atezolizumab. Finally, we attempted to understand whether Glo1/MG axis could represent a possible novel target to overcome or at least reduce this resistance to PD-L1/PD-1 blockade by atezolizumab.

By using mPCa cell models, we demonstrated that MG, through its major derived AGE, 5-hydro-5-methylimidazolone (MG-H1), is able to reduce PD-L1 expression and sustain lymphocytes viability and functionality. More importantly, Glo1, overexpressed in mPCa cells, by limiting the intracellular level of MG-derived MG-H1, prevents this pathway to take place, leading to immunosuppression. Finally, we found that Glo1/MG axis importantly contributes to the refractoriness of mPCa to the anti-PD-L1 therapy by atezolizumab [38] and that targeting Glo1 significantly improves this response.

## 2. Materials and Methods

### 2.1. Cohort Description, Tissue Samples, and Histopathological Analysis

Formalin-fixed paraffin-embedded radical prostatectomy-derived specimens were collected from 90 patients from the Pathological Archives of the Division of Anatomic Pathology and Histology of the University of Perugia following oral informed consent and in accordance with the Declaration of Helsinki and local ethics committee approval (n. 2019-30) between 2005 to 2006. Only the cases with a complete clinical follow-up until 30 June 2017 were considered.

Prostate cancers were re-classified, using the original hematoxylin and eosin sections, in five grade groups, from 1 to 5, according to both the WHO-ISUP 2016 [39] and Epstein’s [40] classifications. After that, the neoplasms were grouped according to cancer aggressiveness into low grade (LG), encompassing the cases within the Grade Group 1 and 2, and a high grade (HG), which included the cases belonging to Grade Group from 3 to 5. The hematoxylin and eosin sections also allowed to analyze the tumor infiltrating lymphocytes (TILs), assessing both the localization (glandular, peri-glandular and stromal), the extent (focal, multifocal and diffuse), and the grade of the TILs (1 = mild, 2 = moderate and 3 = severe), adapting the Nickel et al. histological classification of chronic prostatitis [41].

Four-micrometer-thick formalin-fixed paraffin-embedded sections of the selected prostate cancers were immunohistochemically stained (IHC) for Glo1 (dilution 1:4500, GeneTex, cat. # GTX15747, Prodotti Gianni Srl, Milan, Italy) and PD-L1 (dilution 1:200, Cell Signaling Technology, cat. # 13684, Euroclone, Milan, Italy), with a heat-induced antigen retrieval of 20 min at pH 6 followed by incubation of primary antibody for 15 min and a heat-induced antigen retrieval of 20 min at pH 9 followed by incubation of primary antibody for 15 min, respectively. Together with Glo1 and PD-L1 expression, perineural invasion, tumor extension (T), and lymph nodes status (N) were assessed according to the 8th Ed. of the American Joint Committee on Cancer (AJCC). IHC was performed on a BOND-III fully automated immunohistochemistry stainer (Leica Biosystems, Wetzlar, Germany). A trained pathologist (EP) semi-quantitatively evaluated the immunostains. Each slide was independently assessed by a second trained pathologist (MM) who was blinded to diagnostic and clinical information associated with the cases in the study.

The PD-L1 expression was considered positive when ≥ 1% of the tumoral cells presented a cellular membrane immunolabeling. The macrophages, the lymphocytes, and the nervous structure were used as positive internal controls of the immunostains.

Regarding Glo1, the intensity of the mean cytoplasmatic immunostaining of tumoral cells was considered 0 = absent, 1 = mild, 2 = moderate, 3 = strong and three classes of expression were obtained: the negative (Glo1 intensity 0), the low expression (encompassing the Glo1 intensity class 1), and high expression class (Glo1 intensity from 2 to 3). The label of luminal cell of normal prostate glands was used as positive internal control. 

### 2.2. Reagents

Methylglyoxal (MG) and aminoguanidine bicarbonate (AG) were purchased from Merck Spa (Milan, Italy). We excluded the presence of significant formaldehyde contamination as previously described [42]. Recombinant human interferon gamma (IFN-γ) was purchased from Sigma-Aldrich (Milan, Italy). The bicinchoninic acid (BCA) kit for protein quantification was from Thermo Fisher Scientific (Monza, Italy). Atezolizumab (anti-PD-L1) was from Selleckchem (Divibio Science Italia, Verona, Italy). All the other reagents, where not otherwise specified, were acquired from Sigma-Aldrich (Milan, Italy).

### 2.3. Cell Models and Culture Conditions

DU145 and PC3 prostate cancer cell lines were purchased by the American Type Culture Collection (ATCC) and cultured in RPMI medium at 37 °C and 5% CO_2_ [25]. Primary CD8+ cytotoxic T cells were from ATCC and cultured in RPMI medium at 37 °C and 5% CO_2_ as per the suppliers’ recommendations.

### 2.4. Cell Co-Cultures

Human primary CD8+ cytotoxic T cells were plated at a density of 2 × 10^6^ per well in 6-well plates and stimulated with anti-CD3 (10 µg/mL) and anti-CD28 (2 µg/mL) monoclonal antibodies (DBA Italia Srl, Segrate, Italy) for 48 h to promote T-cell activation [13]. Then CD8+ were co-cultured with PCa cells at a 10:1 ratio for 16 h at 37 °C and subsequently harvested and used for further analyses.

### 2.5. RNA Extraction, Reverse Transcription, and Real-Time Reverse Transcriptase-Polymerase Chain Reaction (RT-PCR)

Total cellular RNA was extracted with TRIzol Reagent (ThermoFisher Scientific, Milan, Italy). cDNA synthesis was performed using 1 µg of RNA with the RevertAid™ H Minus First Strand cDNA Synthesis Kit (ThermoFisher Scientific, Milan, Italy). The expression of genes of interest (versus β-actin) was evaluated by RT-PCR using the MX3000P Real-Time PCR System (Agilent Technology, Milan, Italy). The sequences of the oligonucleotide primers are: Glo1, sense 5′-CTCTCCAGAAAAGCTACACTTTGAG -3′ and antisense 5′-CGAGGGTCTGAATTGCCATTG-3′; PD-L1, sense 5′-GCTATGGTGGTGCCGACTAC-3′ and antisense 5′-TTGGTGGTGGTGGTCTTACC3′; β-actin, sense 5′-CACTCTTCCAGCCTTCCTTCC-3′ and antisense 5′-ACAGCACTGTGTTGGCGTAC-3′. PCR reactions were performed in a total volume of 20 µL, which contained 25 ng of cDNA, 1X Brilliant II SYBR^®^ Green QPCR Master Mix, ROX Reference Dye, and 600 nM of specific primers. The thermal cycling conditions were: 95 °C for 5 min (1 cycle), 95 °C for 20 s, and 60 °C for 30 s (45 cycles). Possible co-amplification of unspecific targets was verified by performing melting curves for each primer pair in standard conditions. Comparative analysis of gene expression was obtained by the 2−(∆∆CT) method [42].

### 2.6. Preparation of Cell Lysates for SDS-PAGE

Sub-confluent cells were lysed with pre-cooled radio-immunoprecipitation assay (RIPA) lysis buffer (ThermoFisher Scientific, Milan, Italy) containing Halt Protease Inhibitor Cocktail and Halt Phosphatase Inhibitor Cocktail (ThermoFisher Scientific, Milan, Italy) as per the manufacturer’s instructions. The obtained protein extracts were used for protein or enzyme activity measurements and for the quantification of total protein content by a BCA kit (ThermoFisher Scientific) with bovine serum albumin as a standard [43]. 

### 2.7. Glyoxalase 1 (Glo1) Enzyme Activity and Protein Level

Glo1 activity was assayed as previously described [44] Inizio modulo Fine modulo. Glo1 protein level was detected by a specific GLO1 ELISA Kit (Human) (OKCD08161) (Aurogene, Rome, Italy).

### 2.8. SDS-PAGE and Western Blot 

SDS-PAGE and Western blot were performed as previously described [45,46]. In short, 35 µg of each sample were treated with Laemmli buffer, boiled for 5 min, resolved on a 4–20% SDS-PAGE, and then blotted onto a nitrocellulose membrane by the iBlot Dry Blotting System (ThermoFisher Scientific). Roti-Block (Prodotti Gianni S.r.l., Milan, Italy) was used for 1 h at room temperature to block non-specific binding sites. Membranes were then incubated overnight at 4 °C with anti-Glo1 Ab (dilution 1:1000, GeneTex, cat. # GTX15747, Prodotti Gianni Srl, Milan, Italy) or anti-PD-L1 Ab (dilution 1:200, Cell Signaling Technology, cat. # 13684, Euroclone, Milan) or anti-MG-H1 Ab (dilution 1:1000, Cell Biolabs, cat. # STA-011, DBA Italia Srl, Segrate, Italy). Subsequently, membranes were washed with Tris-buffered saline/Tween, and incubated for 1 h at room temperature with the peroxidase-conjugated anti-mouse secondary antibody (dilution 1:2000, Merck Spa, Milan, Italy). Visualization was carried out by the ECL system (Merck Spa, Milan, Italy). All membranes were subsequently re-probed with the housekeeping Ab anti-β-actin (dilution 1:1000, Santa Cruz Biotechnology, cat. # sc-376421, DBA Italia Srl, Segrate, Italy).

### 2.9. MG-H1 Protein Adducts Detection

MG-H1 protein adducts were also detected by using a competitive enzyme-linked immunosorbent assay (ELISA) kit (DBA Italia Srl, Segrate, Italy) as per the manufacturer’s instructions.

### 2.10. siRNA Transfection

DU145 and PC3 cells were transiently transfected as previously described [25]. In particular, a pool of four small interfering RNA (siRNA) oligonucleotides-targeting Glo1 (siGlo1) or a pool of non-targeting siRNA oligonucleotides (siCtr) (ONTARGET plus siCONTROL), as a negative control to exclude off-targets, was used (Dharmacon RNA Technologies, Carlo Erba, Milan, Italy). DharmaFECT 2 was the transfection reagent (Dharmacon RNA Technologies, Carlo Erba, Milan, Italy). 

### 2.11. Apoptosis and Cell Viability Detection, Cell Counting

Apoptosis was measured by evaluating caspase-3 activation with a specific human Caspase-3 (active) ELISA kit (Invitrogen, Milan, Italy). Cell viability was evaluated by MTT assay, as previously described [43]. Cell number was evaluated by using a counting chamber under an inverted microscope. 

### 2.12. TNF-α and IL-2 Detection

TNF-α and IL-2 were measured by using specific commercially available ELISA kits from Abcam (Prodotti Gianni Spa, Milan, Italy) according to the manufacturer’s instructions.

### 2.13. Statistical Analysis

Categorical variables were compared between groups using Chi-square test or Fisher’s exact test as appropriate. Correlation analyses were carried out with the Spearman’s correlation test. Results, from three independent experiments, were expressed as means ± standard deviation (SD). One-way analysis of variance with Dunnett’ s correction was employed to determine differences among groups. Statistical significance was set at *p* < 0.05.

## 3. Results

### 3.1. Patients Clinico-Pathological Characteristics and TILs Analysis

To reach the objective of our research, we first performed an IHC study on a population of 90 patients whose clinical-pathological characteristics are presented in Table 1.

The median age was 66 years (range 51–76). Twenty-seven (30%) patients died at the end of the observational period, 12 (44%) of which from PCa and 5 (19%) with PCa. The median follow-up period was of 11 years, with a median of 5 years for the patients who died from PCa, 7 years for both patients who died with PCa and patients who died from another cause, 12 years for alive patients without disease. Moreover, 17 (19%) cases relapsed.

Regarding the grade groups, 26 (29%) of the cases belonged to the Group 1, 20 (22%) to the Group 2, 18 (20%) to the Group 3, 10 (11%) to the Groups 4, and 16 (18%) to the Group 5. Moreover, at the diagnosis, most of the tumors (73; 81%) were confined to the prostate, 65 (72%) presented a perineural invasion, and 8 (10%) locoregional lymph node metastasis.

Concerning the TILs localization, most of the cases (50; 56%) showed a peri-glandular arrangement of the lymphocytic infiltrates and no one showed a glandular one. As regards the extent, most of the neoplasms had either a focal (52, 58%) or a multifocal (31; 34%) extension of the TILs. Finally, 60 (67%) of the cases showed a mild grade of the TILs.

### 3.2. PD-L1 IHC Analysis and Clinical-Pathological Associations

As shown in Table 1, 9 (10%) PCa displayed a PD-L1 expression on the tumor cells. Among these cases, 6 (67%) patients were alive without disease, 2 (22%) died from PCa, and 1 (11%) died for another cause whit still ongoing PCa. Most of the PD-L1 positive cases (8; 89%) showed, also, a perineural invasion. As regards the localization of the TILs among the cancers with tumor expression of PD-L1, all the 9 (100%) cases showed a peri-glandular localization of the TILs; among this last group, 5 (56%) and 4 (44%) presented a focal or multifocal extension, respectively. Moreover, the majority of tumor which expressed PD-L1 (5; 56%), showed, simultaneously, a moderate grade of TILs.

### 3.3. Glo1 IHC Analysis and Clinical-Pathological Associations

As presented in Table 1, 75 (83%) cases were positive for Glo1 in tumoral cells, 49 of which (54%) were included in the high expression class of the protein. Of note, 9 (75%) of the patients who died from PCa showed a high expression of Glo1, highlighting a trend of worse outcomes for the neoplasms expressing high levels of this protein (*p* = 0.094). Moreover, high expression of Glo1 was statistically associated with both the high aggressiveness group of diseases, the perineural invasion, an extra-prostatic extension of the disease, lymph nodes metastasis, and relapse (respectively, 35, 80%, *p* < 0.01; 43, 66%, *p* < 0.01; 14, 82%, *p* = 0.021; 8, 100%, *p* = 0.025 and 14, 82%, *p* = 0.023). No statistically significant associations were found between Glo1 expression and patients’ age. 

As regards the associations of Glo1 and TILs, there were no statistically significant relationships between the classes of expression of Glo1 and the localizations of the TILs. However, about half of the cases (26; 52%) with a high expression of Glo1 showed a simultaneous peri-glandular localization of TILs. As regards both the extent and the grade of TILs, although all the cases (2, 100% and 3, 100%, respectively) with either diffuse or high-grade TILs presented a high expression of Glo1, there were not statistically significant associations between these parameters. 

Interestingly, we observed that 7 out of 9 patients (78%) with the highest expression of Glo1 showed simultaneously also the highest expression of PD-L1. Moreover, of these 7 patients, 2 (22%) developed liver and bone metastases and died from PCa. Finally, and of note, these two patients presented a peri-glandular localization and a focal extension of the lymphocyte infiltrate (TILs), of moderate to mild grade (Figure 1).

Altogether these observations suggested a possible synergic role between Glo1 and PD-L1 in negatively modulating the lymphocyte infiltrate in aggressive PCa and potentially a novel mechanism, based on Glo1/PD-L1 axis, contributing to PCa immune escape.

### 3.4. Glo1 and PD-L1 Expression in Biopsies from mPCa Patients 

To further deepen the possible association between Glo1 and PD-L1 in mPCa, we also evaluated, at mRNA and protein level, the expression of both molecules in biopsy samples obtained from metastatic (*n* = 30, stage M1) and non-metastatic PCa (*n* = 30 of which 15 stage pT2, cancer confined to the prostate and 15 stage pT3, extra-prostatic extension and/or seminal vesicle involvement) patients, whose characteristics were previously described [25]. Briefly, mean age (years) ± SD was 66.4 ± 6.3 in the non-metastatic PCa and 67.0 ± 7.4 in the metastatic group. Mean PSA levels (ng/mL) ± SD were 11.4 ± 4.6 or 586.4 ± 254.0 in the non-metastatic and metastatic PCa group, respectively. Median Gleason score was 6 for the non-metastatic group and 9 for the metastatic cohort.

We observed that patients with metastatic (M) PCa showed significantly higher levels of both Glo1 (Figure 2a) and PD-L1 (Figure 2b) expression with respect to patients with non-metastatic (NM) PCa. Moreover, a positive significant correlation was observed between Glo1 and PD-L1 (Spearman’s correlation coefficient = 0.87, *p* = 0.001) in the metastatic group.

Since Glo1 is an enzymatic protein involved in MG removal, we also measured Glo1 specific activity as well as MG levels, through its major derived AGE, MG-H1. In line with the trend in mRNA and protein levels, also Glo1 enzyme activity was significantly higher in the metastatic compared to the non-metastatic group (Figure 2c). As expected, MG-H1 amounts were lower in the metastatic compared with the non-metastatic group (Figure 2c). Overall, these data reinforced the hypothesis suggested by the IHC analysis of a link between Glo1 and PD-L1 in mPCa, and indicated a possible role of MG-H1 in this potential Glo1/PD-L1 axis.

### 3.5. Glo1 Sustains PD-L1 Expression through the Negative Control of MG-H1 in mPCa Cells

To demonstrate that Glo1 might causatively control PD-L1 expression through MG-H1 in mPCa cells, we used DU145 and PC3 cell lines, models of metastatic and castration-resistant PCa [47,48,49].

First, we exposed both cell lines to the non-toxic (Appendix A) 10 µM MG concentration for 48 h and found that it induced a significant increase in MG-H1 intracellular levels, evaluated either by Western blot (Figure 3a) or ELISA (Figure 3b). In parallel, MG induced a marked decrease in PD-L1 expression both at transcript (Figure 3c) and protein level; the latter was evaluated in immunoblot by using the same Ab as in IHC analyses (Figure 3d). PD-L1 protein expression was also measured by ELISA (Figure 3e). Since for both MG-H1 and PD-L1, results from Western blot overlapped with those from ELISA, subsequent analyses were performed only using this latter method. 

Altogether, these results suggested that MG might decrease PD-L1 expression through MG-H1, in both cell lines. In order to show this, we pre-treated DU145 and PC3 cells with AG, a scavenger of MG [50] and evaluated MG-H1 and PD-L1 levels, upon or not MG exposure. We found that AG significantly reverted MG-induced MG-H1 intracellular content (Figure 4a) and brought back PD-L1 transcript and protein levels (Figure 4b) to those of control, proving that MG inhibits PD-L1 expression through MG-H1 generation in mPCa cells.

In order to show that MG-driven MG-H1-mediated PD-L1 inhibition was under Glo1 control, we first silenced Glo1 (Appendix A) and subsequently exposed cells to MG upon Glo1 silencing. After that, we measured both MG-H1 and PD-L1 levels. Compared with siCtr cells, siGlo1 treatment increased MG-H1 intracellular levels (Figure 4c) and decreased PD-L1 expression (Figure 4d) in both DU145 and PC3 cells. More importantly, these changes were potentiated upon MG treatment, thus supporting a mechanism whereby Glo1 depletion induces MG-H1 intracellular accumulation that in turn decreases PD-L1 expression. In other words, these results confirmed that Glo1 sustains PD-L1 expression by suppressing intracellular levels of MG-H1 in metastatic PCa cells.

### 3.6. Glo1/MG-H1/PD-L1 Axis Is Involved in mPCa Immunosurveillance Escape

It is well-known that for a cancer to metastasize it has to escape anti-tumor immune response, especially CD8+ cytotoxic T-cell-mediated elimination [51,52]. Moreover, it has been demonstrated that the PD-L1/PD-1 pathway inhibits the anti-tumor immune response of T cells [13] and that PD-L1 preferentially modulates the secretion of regulatory cytokines in the tumor microenvironment [53]. Taking into account all this, we examined whether the decreased PD-L1 expression, determined by MG-H1 accumulation through Glo1 silencing, in mPCa cells affected the tumor microenvironment, in terms of T-cell response, evaluated either through apoptosis and cell number or cytokine release. To this aim, we performed co-culture experiments between DU145 and PC3 cells with CD8^+^ T cytotoxic cells. In particular, we demonstrated that siGlo1-treated mPCa cells, characterized by a low PD-L1 expression (Figure 4), were able to decrease PD-1+/CD8+ T-cell apoptosis compared with siCtr-treated mPCa cells, as shown by the decrease of active caspase-3 (Figure 5a), the final effector of caspase-dependent apoptosis. Consistently, CD8+ cell number was significantly higher upon co-cultures with siGlo1-treated mPCa cells compared with that upon co-cultures with siCtr-treated mPCa cells (Figure 5b). 

As to cytokine release by CD8+ cells, we focused on TNF-α and IL-2, cytokines usually associated with cancer eradication [13]. We found that upon co-culturing with siGlo1-treated mPCa cells, CD8+ cells increased TNF-α (Figure 5c,d) and IL-2 (Figure 5e,f) secretion in the culture media, compared with co-culturing with siCtr-treated mPCa cells.

Collectively these data, indicated that Glo1 silencing affects CD8+ T-cell response by sustaining their viability and functionality. This means that in mPCa cells Glo1, by negatively controlling CD8+ viability and functionality, is involved in immunosurveillance escape.

Induction of PD-L1 by 5 ng/mL IFN-γ for 8 h (Appendix A) [54,55], upon siGlo1 silencing, reverted Glo1 knock-down-mediated apoptosis (Figure 6a) and cell number (Figure 6b) as well as cytokine secretion (Figure 6c–f), thus suggesting that Glo1 negatively modulates CD8+ cell viability and cytokine secretion through PD-L1-dependent pathway.

Collectively, these data suggest that Glo1, by limiting MG-H1 intracellular level, sustains PD-L1 expression in mPCa cells, keeping active this inhibitory check-point in impairing CD8+ T-cells viability and functionality. Thus, Glo1 represents a novel actor in the scene of mPCa immunosuppressive tumor microenvironment and a novel player in the process of metastatic prostate cancer immunosurveillance escape.

### 3.7. Glo1/MG-H1/PD-L1 Axis Contributes to the Refractoriness of mPCa to the Anti-PD-L1 Therapy by Atezolizumab and Targeting Glo1 Induces Aatezolizumab Response in mPCa Cells

Atezolizumab (Atz) is a humanized engineered IgG1 monoclonal antibody that selectively targets PD-L1, blocking its receptor (PD-1) interactions, which can enhance T-cell responses and improve antitumor activity [56,57,58]. The few recent studies investigating the anticancer effect of this anti-PD-L1 agent in mPCa have preliminarily observed that in heavily pretreated patients, Atz monotherapy demonstrated only a minimal evidence of disease control. This limited efficacy suggests that a combination approach may be needed [59,60] and that additional mechanisms underpinning this low response must exist. To investigate whether Glo1/MG-H1/PD-L1 axis could represent a novel pathway involved in Atz resistance in mPCa, we exposed DU-145 and PC3 cells to the non-toxic (Appendix A) concentration of 10 µg/mL Atz and control IgG isotype [61] and evaluated CD8+ apoptosis and cell number in co-culture experiments. We found that treatment with PD-L1 inhibitor alone still induced a significant increase in CD8+ apoptosis (Figure 7a) and decrease in CD8+ cell number (Figure 7b) compared with IgG isotype or untreated cells, proving its inefficacy in resuming CD8+ number and, potentially, their anticancer defense response. Indeed, in parallel, DU-145 and PC3 cells apoptosis did not increase by Atz treatment compared with control (Figure 7c). Interestingly, Atz significantly increased Glo1 expression and activity (Figure 7d) and, concomitantly, reduced MG-H1 levels (Figure 7e) while increasing PD-L1 expression (Figure 7f) in both DU-145 and PC3 cells.

Altogether these results suggested that Atz instead of boosting CD8+-mediated immune response against mPCa cells, blocked it, through Glo1/MG-H1/PD-L1 axis empowerment. In other words, Atz fueled Glo1/MG-H1/PD-L1 axis that contributed to negatively modulate T-cell anticancer response more efficiently than it did in blocking PD-L1, thus helping explaining, at least in part, the refractoriness of mPCa cells to this anti-PD-L1 therapy. These findings suggested also that Glo1/MG-H1/PD-L1 axis might represent a novel pathway to be targeted to induce Atz efficacy. In fact, upon Glo1 silencing, Atz was markedly effective in killing mPCa cells, as shown by the marked cytotoxic effect on DU-145 and PC3 cells (Figure 8a), in a mechanism involving CD8+ response (decreased apoptosis and increased number) (Figure 8b,c).

## 4. Discussion

Cancer development and progression is a multistep process involving several essential distinctive traits including evasion from immune destruction [62], frequently occurring by cytotoxic immune cells apoptosis [63]. PD-L1/PD-1 axis plays a crucial role in inducing apoptosis of cytotoxic T cells, thus maintaining an immunosuppressive microenvironment [64] that contributes to nurture tumor growth and progression. Hence, antibodies targeting PD-L1 or PD-1 have proved effective in some solid cancers [65,66]. In PCa there are not equivalent effects, which suggests that our understanding on PD-L1/PD-1 axis in this neoplasia is still limited.

In this study, we showed, for the first time, that Glo1, by limiting MG-H1 intracellular accumulation, prevents this specific MG-derived AGE to down-regulate PD-L1 expression in advanced metastatic cell models, all this resulting in PD-L1-mediated inhibition of cytotoxic CD8+ T cells, and consequent promotion of tumor immune evasion, with concomitant cancer progression (Figure 9).

In particular, Glo1-driven PD-L1-mediated inhibition of cytotoxic CD8+ T cells occurred through induction of apoptosis, paralleled, as expected, by a significant reduced number of these T cells, and decreased production of specific anti-cancer cytokines. Our results further improve our general knowledge about the regulatory mechanisms of the PD-L1/PD-1 axis. In fact, although it is one of the most studied immune checkpoints, PD-L1/PD-1 pathway still presents several aspects of its biology that must be clarified [20] and only little is known about signaling pathways regulating PD-L1 expression in tumor cells [67]. In our observed mechanism, a key role in modulating PD-L1 expression in mPCa cells is played by MG-derived MG-H1, that is able to inhibit the expression of this membrane protein, either alone or, as expected, even more, in combination with Glo silencing, thus sustaining CD8+ viability and functionality. How MG-H1 can inhibit PD-L1 expression remains to be elucidated. As known, dicarbonyl adducts derived from MG can exert complex and multiple effects on several biological processes, including regulation of protein stability through their post-translational modifications [68] and/or generation of reactive oxidative species (ROS/RNS) and oxidative stress [69]. Moreover, MG may act as a signal molecule per se [70]. Hence, we hypothesize that MG could inhibit PD-L1 expression through one or more of these pathways. Besides, as an example, although the interplay between ROS and PD-L1 expression is very complex, it has been reported that ROS can engender the down-regulation of PD-L1 expression in cancer cells [71], thus rendering plausible our assumption. Whatever the case may be our results add further knowledge to the immunomodulatory role of MG [35,72], through PD-L1 expression control.

In addition, our results, could, at least in part, help to explain why mPCa is immunologically “cold” and predominantly resistant to immune checkpoint therapy due to fewer tumor-infiltrating T cells [73]. In fact, Glo1 overexpression would sustain PD-L1 expression thus limiting the T-cell number in the microenvironment. In particular, we focused on CD8+ cytotoxic T cells that are considered to be among the key players in the immune elimination of tumors. Moreover, the mechanisms by which malignant cells evade attack by tumor-specific CD8+ T cells are important components of the immune escape process [51]. Hence, our data provide an important contribution to the different immune escape routes till now known [51].

More importantly, Glo1/MG-H1/PD-L1 axis could represent, through up-stream Glo1 inhibition, a novel attractive target for therapeutic strategies aimed at driving PD-1/PD-L1 blockade and/or helping to overcome hurdles of targeting it with already available PD-1 and/or PD-L1 antibodies, such as atezolizumab in this study, which is an urgent need, especially in the advanced stages of PCa, whose outcome is lethal. In fact, results from PCa studies have so far proven mostly disappointing.

Of note, in consideration of the growing importance of PD-L1/PD-1 axis as a target in anti-cancer immunotherapy and of the awareness of its failure in some solid malignancies, this research sets the stage for additional studies aimed at investigating Glo1/MG-H1/PD-L1 pathway in immunotherapy responses and malignant behaviors in broader cancer contexts. Besides, while preparing the draft of the present manuscript, a research reported that stable Glo1 knock-down in DU145 PCa cells and human malignant melanoma was paralleled by pronounced PD-L1 expression [29], thus not only supporting our data but also confirming our perspective of a causative role of Glo1 in controlling PD-L1 expression in other neoplasms.

Importantly, we observed that concomitant Glo1/PD-L1 expression in tissues of patients bearing aggressive and metastatic PCa was associated with a scanty lymphocyte infiltrate (TILs) and poor prognosis. Although this result needs to be confirmed in a larger sample, it would strongly support, together with our in vitro mechanistic model, the clinical relevance of Glo1/PD-L1/TILs crosstalk in PCa progression through immune surveillance elusion and a role for Glo1 and PD-L1 co-expression to predict patients’ prognosis and response to treatment.

As to the prognostic aspect, over the past years has emerged the urgent need for the identification of reliable prognostic biomarkers able to potentially identify mPCa patients. Here, we confirm the role of Glo1 as a marker of PCa progression and dismal prognosis [24,74] and highlight how high expression of Glo1 together with PD-L1 might better function as markers of PCa progression and mortality. Besides, it is well acceptable that the combination of more markers in each single case, with respect to a single one, may be a more predictive factor for the association to a disease progression.

As to the aspect associated with Glo1 and PD-L1 co-expression to predict treatment effect, we speculate that careful patient selection (patients with positive Glo1 and PD-L1 tumors) is likely to be critical for the success of PD-L1-targeting inhibitors in particular, the one used in the present research, atezolizumab. Besides, at present, specific biomarkers that can be useful to select potentially responsive patients to immunotherapeutic agents are missing. In this context, prospectively, Glo1/PD-L1 axis could serve as biomarker for selecting appropriate therapies for every single patient (in the context of the so-called “precision medicine”) and, hopefully, help to respond to several unsolved questions such as timing of immunotherapies, possibility of withdrawing therapy and rechallenged after progression.

In the end, our data provide a molecular basis to additional studies aimed at investigating the role of Glo1/PD-L1 as relevant biomarkers to predict treatment response and, possibly, to improve the clinical response rate of PD-L1/PD-1 blockade in aggressive mPCa, and maybe across other cancer types, where Glo1 is overexpressed to reduce MG-H1 levels. 

## 5. Conclusions

In summary, we report a previously unidentified mPCa cell-intrinsic mechanism, based on the Glo1/MG-H1/PD-L1 axis, by which cancerous cells elude CD8+ T-cell-mediated immune surveillance toward cancer progression and survival, thus suggesting that down-regulating Glo1 may become a promising therapy that, targeting tumor immune evasion, inhibits malignant growth of mPCa cells. Ongoing and future studies will further investigate biologic consequences of modulating this axis in vivo, with the goal of clinical translation. The absence of in vivo studies may represent a limit of the present study.

## Figures and Tables

**Figure 1 cancers-13-02965-f001:**
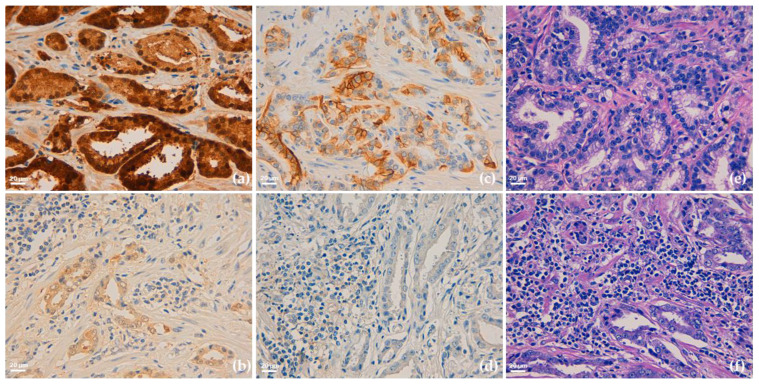
Representative immunohistochemical staining of Glyoxalase 1 (Glo1) (**a**,**b**), PD-L1 (**c**,**d**) expression and tumor infiltrating lymphocytes (TILs) asset (**e**,**f**) in human prostate cancer tissues. Images (**a**,**c**,**e**) refers to the same patient and are characterized by a parallel high expression of both Glo1 (**a**) and PD-L1 (**c**), and an almost absence of stromal TILs. As controls, images (**b**,**d**,**f**) refers to another patient and are characterized by a parallel low expression of both Glo1 (**b**) and PD-L1 (**d**) and a diffuse presence of grade 3 TILs; (**e**,**f**) hematoxylin and eosin staining. Original magnification: 400×. The scale bar indicates 20 µm.

**Figure 2 cancers-13-02965-f002:**
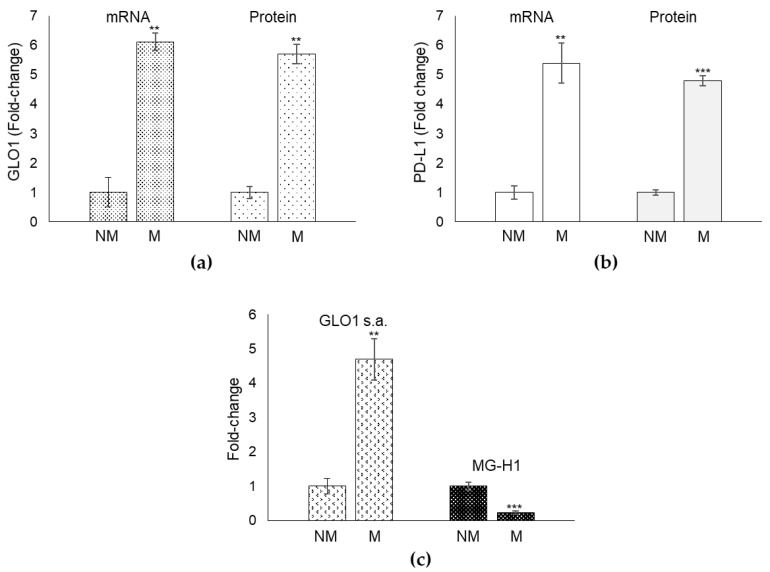
Glyoxalase 1 (Glo1) and PD-L1 mRNA expression in biopsies from metastatic PCa patients. (**a**) Glo1 expression, at mRNA and protein level, and (**b**) PD-L1 expression, at mRNA and protein level, in biopsy samples obtained from metastatic (*n* = 30, stage M1) and non-metastatic PCa (*n* = 30 of which 15 stage pT2, cancer confined to the prostate and 15 stage pT3, extra-prostatic extension and/or seminal vesicle involvement) patients. (**c**) Glo1 enzyme activity was measured in total protein extracts spectrophotometrically, while MG-H1 by a specific ELISA assay. ** *p* < 0.01; *** *p* < 0.001.

**Figure 3 cancers-13-02965-f003:**
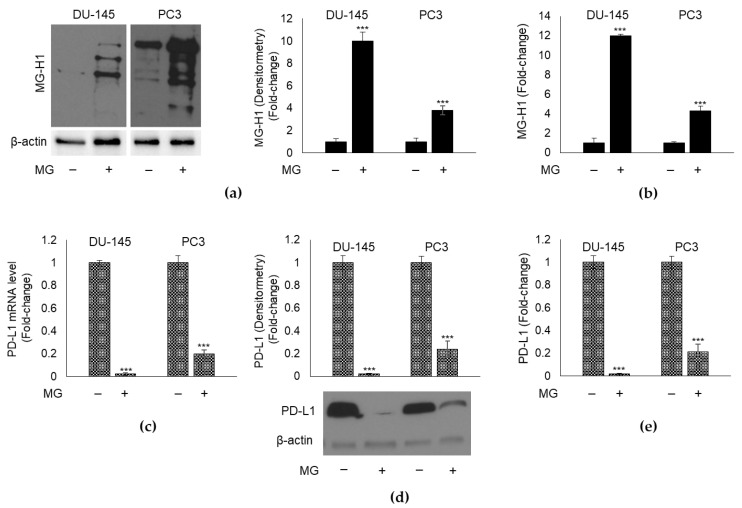
Effect of methylglyoxal (MG) on MG-derived AGE, MG-H1, and PD-L1 expression in mPCa cells. MG, at the concentration of 10 µM, was administrated to DU-145 and PC3 cells for 48 h and MG-H1 levels were measured by Western blot (**a**) and a specific ELISA kit (**b**). The expression of PD-L1 was evaluated at (**c**) mRNA level by RT-qPCR, and protein level by (**d**) Western blot and (**e**) a specific ELISA kit. The molecular weight markers associated with the blots are reported in Appendix A. The histograms indicate mean ± SD of three different cultures, and each was tested in duplicate. *** *p* < 0.001 significantly different from untreated control cells.

**Figure 4 cancers-13-02965-f004:**
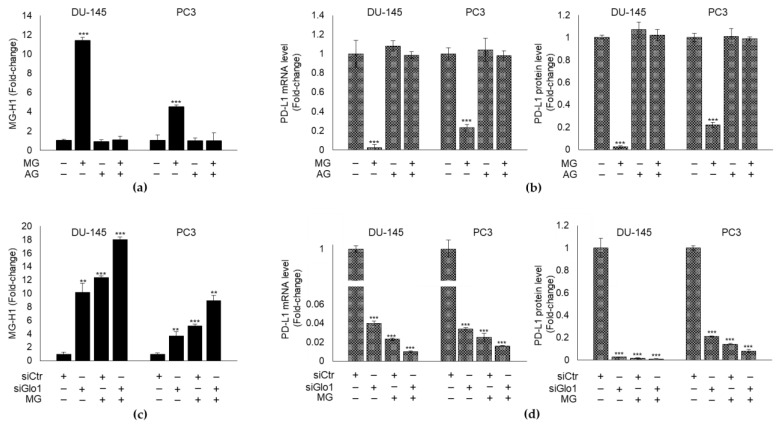
Glyoxalase 1 (Glo1) sustains PD-L1 expression by limiting MG-H1 intracellular accumulation in mPCa cells. DU-145 and PC3 mPCa cells were exposed to 10 µM MG for 48 h after a pre-treatment with 1 mM aminoguanidine (AG) for 48 h and (**a**) MG-H1 intracellular levels were measured by a specific ELISA kit, while (**b**) PD-L1 expression was evaluated at mRNA level and protein level by RT-qPCR and a specific ELISA kit, respectively. The effect of Glo1 silencing (siGlo1) or siCtr (negative control) was evaluated on (**c**) MG-H1 and (**d**) PD-L1 levels measured as above. The histograms indicate mean ± SD of three different cultures, and each was tested in duplicate. (−) untreated and (+) treated cells. ** *p* < 0.01; *** *p* < 0.001 significantly different from control cells.

**Figure 5 cancers-13-02965-f005:**
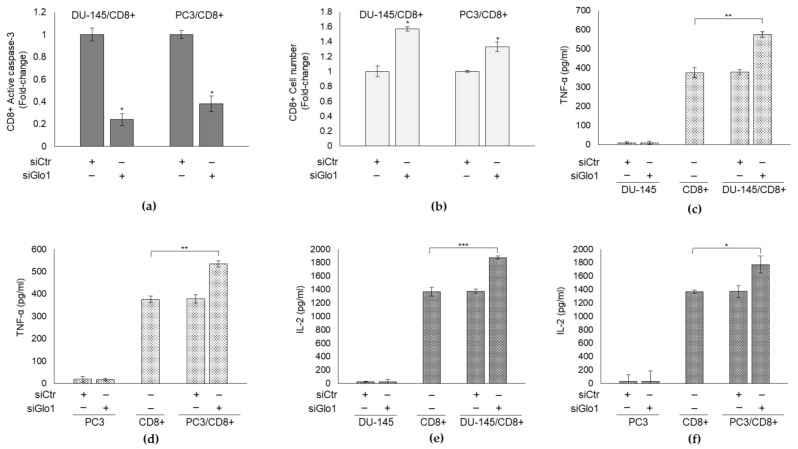
Glyoxalase 1 (Glo1) is involved in mPCa immunosurveillance escape. DU145 and PC3 cells transiently transfected with Glo1 small interfering RNA (siGlo1) or control non-targeting small interfering RNA (siCtr) were co-cultured with CD8+ cytotoxic T cells and (**a**) CD8+ apoptosis was evaluated by active caspase-3 expression, with a specific ELISA kit, (**b**) CD8+ cell number was measured by cell counting, (**c**,**d**) TNF-α and (**e**,**f**) IL-2 secretion by CD8+ cells in the culture medium was detected by specific ELISA kit. The histograms indicate mean ± SD of three different cultures, and each was tested in duplicate. (−) untreated and (+) treated cells. * *p* < 0.05, ** *p* < 0.01; *** *p* < 0.001.

**Figure 6 cancers-13-02965-f006:**
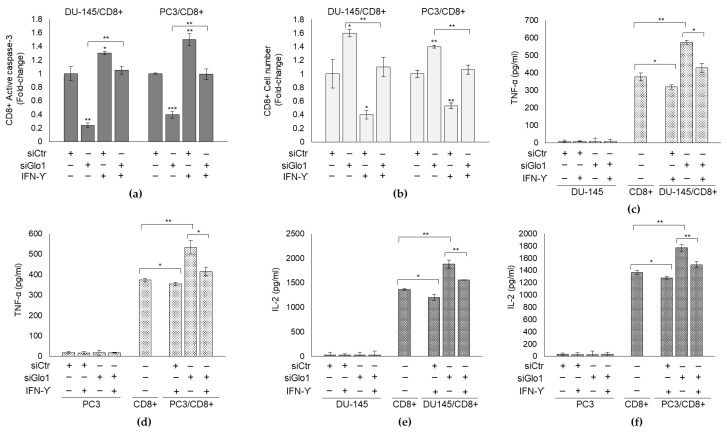
Glyoxalase 1 (Glo1)/MG-H1/PD-L1 axis is involved in mPCa immunosurveillance escape. DU145 and PC3 cells transiently transfected with Glo1 small interfering RNA (siGlo1) or control non-targeting small interfering RNA (siCtr) were co-cultured with CD8+ cytotoxic T cells and exposed to 5 ng/mL IFN-γ for 8 h. (**a**) CD8+ apoptosis was evaluated by active caspase-3 expression, with a specific ELISA kit, (**b**) CD8+ cell number was measured by cell counting, (**c**,**d**) TNF-α and (**e**,**f**) IL-2 secretion by CD8+ cells in the culture medium was detected by specific ELISA kits. The histograms indicate mean ± SD of three different cultures, and each was tested in duplicate. (−) untreated and (+) treated cells. * *p* < 0.05, ** *p* < 0.01; *** *p* < 0.001.

**Figure 7 cancers-13-02965-f007:**
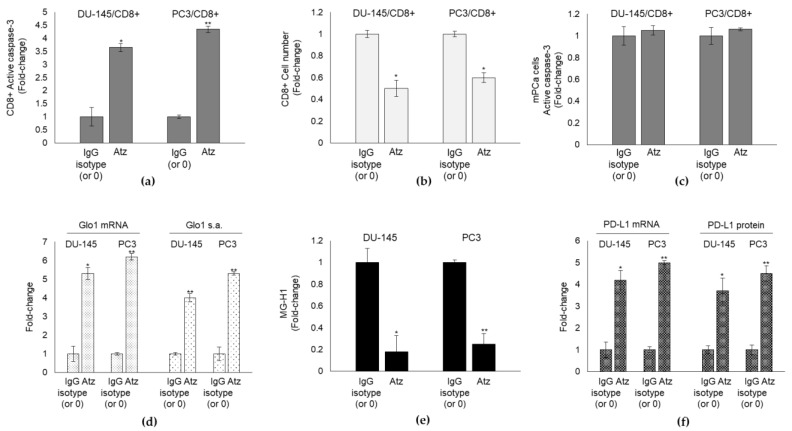
Glyoxalase 1 (Glo1)/MG-H1/PD-L1 axis contributes to the refractoriness of mPCa to the anti-PD-L1 therapy by Atezolizumab (Atz). DU145 and PC3 cells were co-cultured with CD8+ cytotoxic T cells and exposed to 10 µg/mL Atz and control IgG isotype. (**a**) CD8+ apoptosis was evaluated by active caspase-3 expression, with a specific ELISA kit, (**b**) CD8+ cell number was measured by cell counting, (**c**) DU-145 and PC3 cell apoptosis was evaluated by measuring active caspase-3 expression by a specific ELISA kit, (**d**) Glo1 mRNA levels, and specific activity were evaluated by RT-qPCR and spectrophotometrically; (**e**) MG-H1 intracellular levels were measured by a specific ELISA kit, while (**f**) PD-L1 expression was evaluated at mRNA level and protein level by RT-qPCR and a specific ELISA kit, respectively. The histograms indicate mean ± SD of three different cultures, and each was tested in duplicate. * *p* < 0.05, ** *p* < 0.01.

**Figure 8 cancers-13-02965-f008:**
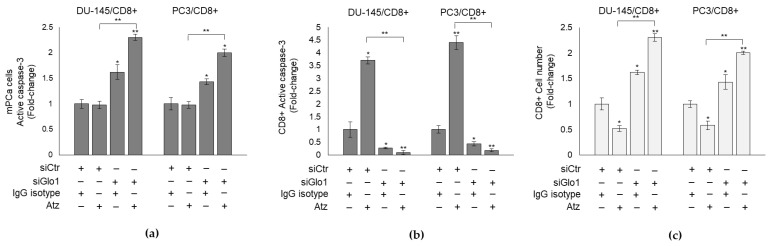
Targeting Glo1 induces atezolizumab (Atz) response in mPCa cells. siGlo1-silenced DU145 and PC3 or controls (siCtr) cells were co-cultured with CD8+ cytotoxic T cells and exposed to 10 µg/mL Atz or control IgG isotype. (**a**) mPCa cells apoptosis or (**b**) CD8+ apoptosis was evaluated by active caspase-3 expression, with a specific ELISA kit; (**c**) CD8+ cell number was measured by cell counting. The histograms indicate mean ± SD of three different cultures, and each was tested in duplicate. * *p* < 0.05, ** *p* < 0.01.

**Figure 9 cancers-13-02965-f009:**
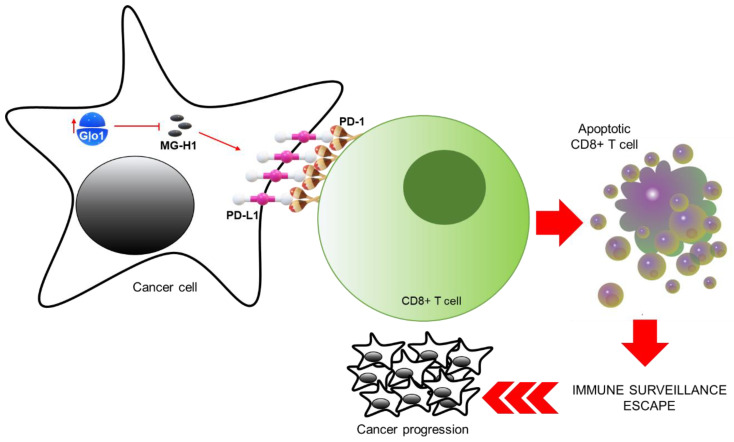
Glyoxalase 1-dependent methylglyoxal depletion sustains PD-L1 expression in metastatic prostate cancer cells: a novel mechanism in cancer immunosurveillance escape. Glo1, by limiting MG-H1 intracellular accumulation, prevents this specific MG-derived AGE to down-regulate PD-L1 expression in advanced metastatic cell models, all this resulting in PD-L1-mediated inhibition of cytotoxic CD8+ T cells, and consequent promotion of tumor immune evasion, with concomitant cancer progression.

**Table 1 cancers-13-02965-t001:** Clinical-pathological parameters, Glo1 and PD-L1 expression, tumor infiltrating lymphocytes (TILs) asset.

Parameter	Glo1 Negative		Glo1 Low		Glo1 High			Total (Row)	
	N	%	N	%	N	%	*p*	N	%
	15	17	26	29	49	54		90	100
**Age**									
<66 yrs	10	25	11	27	19	48	0.158	40	44
≥66 yrs	5	10	15	30	30	60		50	56
**Exitus**									
DfD	2	17	1	8	9	75	0.094	12	13
DwD	0	0	0	0	5	100		5	6
DaC	0	0	5	50	5	50		10	11
AwD	13	20	20	32	30	48		63	70
**Relapse**									
No	13	18	25	34	35	48	0.023	73	81
Yes	2	12	1	6	14	82		17	19
**Grade Groups**									
LG	13	28	19	41	14	31	<0.01	46	51
HG	2	4	7	16	35	80		44	49
**Perineural Invasion**									
No	9	36	10	40	6	24	<0.01	25	28
Yes	6	9	16	25	43	66		65	72
**pT**									
T2	15	20	23	32	35	48	0.021	73	81
T3	0	0	3	18	14	82		17	19
**pN**									
N−	13	17	25	34	36	49	0.025	74	90
N+	0	0	0	0	8	100		8	10
**TILs localization**									
P	9	18	15	30	26	52	0.104	50	56
S	3	8	10	29	22	63		35	39
A	3	60	1	20	1	20		5	5
**TILs extent**									
F	8	15	16	31	28	54	0.300	52	58
M	4	13	9	29	18	58		31	34
D	0	0	0	0	2	100		2	2
A	3	60	1	20	1	20		5	6
**TILs grade**									
1	8	13	19	32	33	55	0.227	60	67
2	4	18	6	27	12	55		22	24
3	0	0	0	0	3	100		3	3
A	3	60	1	20	1	20		5	6
**PD-L1**									
Positive	0	0	2	22	7	78	0.278	9	10
Negative	15	18	24	30	42	52		81	90

DfD = died from PCa; DwD = died with PCa; DaC = died from another cause; AwD = alive without disease. LG = low grade group of cancer aggressiveness; HG = high grade group of cancer aggressiveness. T2 = PCa confined to prostate; T3 = extra-prostatic extension of PCa. N− = no locoregional lymph node metastasis; N+ = locoregional lymph node metastasis; P = peri-glandular; S = stromal; A = absent; F = focal; M = multifocal; D = diffuse; 1 = mild; 2 = moderate; 3 = severe.

## Data Availability

The data presented in this study are available on request from the corresponding author.

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
