# Peer review of "Glyoxalase-1-Dependent Methylglyoxal Depletion Sustains PD-L1 Expression in Metastatic Prostate Cancer Cells: A Novel Mechanism in Cancer Immunosurveillance Escape and a Potential Novel Target to Overcome PD-L1 Blockade Resistance"

_cancers, 2021, doi:10.3390/cancers13122965_

Round 1
Reviewer 1 Report
In this study, the authors firstly demonstrated that high level of Glo1 is significantly associated with poor clinical output in their patient cohort (N=90), which is in agreement with other studies showing high expression of Glo1 is risk factor in Prostate Cancer (PCa) . In the in vitro experiments, the authors provided several lines of evidences showing that Glo1-mediated inhibition on MG-H1 can sustain PD-L1 expression and thus confers cancer cells immuno-surveillance escape. Overall, this study provides a novel mechanism of how PCa cells evade the immuno-checkpoint. However, the reviewer also raise some concerns:
Major concerns:
1)It would be more convincing if the authors can provide some representative IHC staining images of Glo1.
2)It is confounding that the std bars are quite small in figure 1, considering that there are 30 samples in each clinical populations. The authors need to provide a more clear data presentation, a box plot would be more appropriate.
3)The knock down of Glo1 should be validated on the mRNA and protein level.
4) In many of the cell line studies (fig4,5,6,7), PC-3 and Du145 behave almost identical. Even though these two lines are very aggressive PCa model lines, do they express similar levels of Glo1.
Minor concerns:
1)The molecular weight markers should be indicated in the immunoblot image in fig2, and the bands that were considered as MG-H1 should be pointed out.
2)There should be more details on the co-culture methods, as the refereed study did not exactly perform the same experiments.
3)Fig4,5 legend title is same, which can be misleading.
Reviewer 2 Report
In this manuscript , the authors demonstrate that
Glyoxalase 1-dependent methylglyoxal depletion sustains PD- 2
L1 expression in metastatic prostate cancer cells and claim this mechanism may be a potential novel target to overcome PD-L1 blockade resistance.
Although these findings were novel and of clinical interesting/importance , there are several major concerns:
- first, many studies showed poor response of prostate cancer to immunotherapy , which may due to prostate cancer a cold tumor, major due to poor immune response in the epithelial-stoma microinvironment, -> so, demonstrate these novel finding in only prostate cancer cell line model is not enough to convince readers, the model of epithelial-stomal interaction and xenograft mouse model were needed !
- the expression of PD-L1 is low in human prostate cancer, could authors discuss more about how to apply this strategy in clinical practice, like how to treatment mCRPC prostate cancers by this concept? any clinical trail support this concept?
Reviewer 3 Report
The paper by Cinzia Antognelli et al. entitled “Glyoxalase 1-dependent methylglyoxal depletion sustains PD-L1 expression in metastatic prostate cancer cells: a novel mechanism in cancer immunosurveillance escape and a potential novel target to overcome PD-L1 blockade resistance” focuses on the molecular mechanisms contributing to keep an immunosuppressive microenvironment associated with tumor progression and resistance to PD-L1 inhibitors in metastatic prostate cancer. This paper is well-organized and grounded in scientific validity. The subject of this study would be of interest for readers of Cancers. I have a few concerns on the present form.
Comments
1. The authors planned this study to clarify the molecular mechanisms contributing to keep an immunosuppressive microenvironment in metastatic prostate cancer based on the fact that prostate cancer was generally resistant to immune checkpoint inhibitors. Considering the purpose of this study, the authors should add a couple of cell lines as a comparison, which were derived from malignancies with sensitivity to immune checkpoint inhibitors. Malignant melanoma and lung cancer are candidates of comparison.
2. The present manuscript lacks in vivo study, which is required to enhance the credibility of the results. The authors are encouraged to add in vivo study to validate their results if available. If not available, the authors are encouraged to refer to the limitations of this study in Discussion.
Reviewer 4 Report
It is a preclinical study on prostate cancer cells and 90 prostate cancer patients. The axis on Glo1/MG-H1/PD-L1 was evaluated. The role of glyoxalase 1 was evaluated on two different PC cell lines (DU145 and PC3). A better representative of the clinical part would have been to select patients with advanced disease (T4) to see really the relationship between aggressiveness and glo1 expression. Analysis of other tumor types would have been also interesting to look at since it seems to be a general abnormal pathway leading to tumor exhaustion. In addition, inhibitors of Glo1 (on the top of silencing Glo1 experiments), if any, would have been great to investigate to see if by blocking glo1, you can further inhibit MG-H1/PD-l1 pathway.
Minor comment
Table 1: DfD; in the legen modified =DxD by DfD= died from PCa
Round 2
Reviewer 1 Report
The authors have addressed all my concerns. The reviewer looks forward to the future in vivo studies and analysis in a larger patient cohort where a significant association of Glo1 and PD-L1 can be observed.
Reviewer 2 Report
About the limitation of this study, the author "only" said one sentence in the conclusion part:The absence of in vivo studies may represent a limit of the present study.
-> could authors add 2-3 sentences regarding the limitation of this study in the discussion part: like .. further in vivo studies to elucidate the immune interaction of epithelial and stroma in the microenvironment is need ... etc
Reviewer 3 Report
The authors have revised their original manuscript partly according to the reviewers’ comments. In some points, the authors decided to keep their contents unchanged, however, their rebuttal seems almost reasonable.
Reviewer 4 Report
corrections are fine